# CaptureSeq: Hybridization-Based Enrichment of *cpn60* Gene Fragments Reveals the Community Structures of Synthetic and Natural Microbial Ecosystems

**DOI:** 10.3390/microorganisms9040816

**Published:** 2021-04-13

**Authors:** Matthew G. Links, Tim J. Dumonceaux, E. Luke McCarthy, Sean M. Hemmingsen, Edward Topp, Jennifer R. Town

**Affiliations:** 1Department of Animal and Poultry Science, University of Saskatchewan, Saskatoon, SK S7N 5A8, Canada; matthew.links@usask.ca (M.G.L); luke@elmonline.ca (E.L.M.); 2Department of Computer Science, University of Saskatchewan, Saskatoon, SK S7N 5A8, Canada; 3Agriculture and Agri-Food Canada, Saskatoon Research and Development Centre, Saskatoon, SK S7N 0X2, Canada; tim.dumonceaux@canada.ca; 4Department of Veterinary Microbiology, University of Saskatchewan, Saskatoon, SK S7N 5B4, Canada; 5National Research Council Canada, Saskatoon, SK S7N 0W9, Canada; sean.hemmingsen@nrc-cnrc.gc.ca; 6Agriculture and Agri-Food Canada, London Research and Development Centre, London, ON N5V 4T3, Canada; ed.topp@canada.ca

**Keywords:** chaperonin-60, microbiome, hybridization, soil microbiota

## Abstract

Background. The molecular profiling of complex microbial communities has become the basis for examining the relationship between the microbiome composition, structure and metabolic functions of those communities. Microbial community structure can be partially assessed with “universal” PCR targeting taxonomic or functional gene markers. Increasingly, shotgun metagenomic DNA sequencing is providing more quantitative insight into microbiomes. However, both amplicon-based and shotgun sequencing approaches have shortcomings that limit the ability to study microbiome dynamics. Methods. We present a novel, amplicon-free, hybridization-based method (CaptureSeq) for profiling complex microbial communities using probes based on the chaperonin-60 gene. Molecular profiles of a commercially available synthetic microbial community standard were compared using CaptureSeq, whole metagenome sequencing, and 16S universal target amplification. Profiles were also generated for natural ecosystems including antibiotic-amended soils, manure storage tanks, and an agricultural reservoir. Results. The CaptureSeq method generated a microbial profile that encompassed all of the bacteria and eukaryotes in the panel with greater reproducibility and more accurate representation of high G/C content microorganisms compared to 16S amplification. In the natural ecosystems, CaptureSeq provided a much greater depth of coverage and sensitivity of detection compared to shotgun sequencing without prior selection. The resulting community profiles provided quantitatively reliable information about all three domains of life (Bacteria, Archaea, and Eukarya) in the different ecosystems. The applications of CaptureSeq will facilitate accurate studies of host-microbiome interactions for environmental, crop, animal and human health. Conclusions: *cpn60*-based hybridization enriched for taxonomically informative DNA sequences from complex mixtures. In synthetic and natural microbial ecosystems, CaptureSeq provided sequences from prokaryotes and eukaryotes simultaneously, with quantitatively reliable read abundances. CaptureSeq provides an alternative to PCR amplification of taxonomic markers with deep community coverage while minimizing amplification biases.

## 1. Introduction

Life on Earth is classified into hierarchical taxonomic lineages that describe all living systems as having descended from a common ancestor along three evolutionary lines. Using ribosomal RNA-encoding gene sequences, Woese and Fox [1] delineated these domains, which are now known as Bacteria, Archaea, and Eukarya [2]. Most complex microbial communities exist as assemblages replete with representatives from each of these domains, the total genomic complement of which is called a microbiome. Understanding microbial community dynamics requires tools to examine the composition of these complex ecosystems. Advancements in DNA sequencing technology have created new opportunities to simplify the profiling of microbial communities from a diverse range of environments. Insights gained through the study of the diversity of microbiomes in soil, water, plant and animal-associated ecosystems have revealed the powerful effects that microbiome composition and structure can have on how these communities function [3]. A comprehensive understanding of the multifaceted relationships between microorganisms and their environment requires the generation of microbial community profiles that reflect, as accurately as possible, the original composition and quantitative structure of the microbial ecosystem under analysis.

In adapting the use of PCR for amplifying a conserved region of 16S rRNA, Weller and Ward provided the first example of microbial profiling [4]. Since then, microbiologists have increasingly embraced such culture-independent methods of identification [5]. PCR-based amplification of 16S rRNA-encoding genes has become the method of choice for determining the composition of bacterial communities in a wide range of ecological niches [6,7]. More recently, Paul Hebert’s proposed DNA barcoding criteria for Eukarya have established standards for what comprises a robust target for phylogenetic profiling [8]. Alternative universal gene markers for 16S [9], *cpn60* [10], *rpo*B [11], *mcr*A [12] and ITS [13] have been used for profiling microorganisms from bacterial, archaeal and eukaryotic domains, although no single amplification is able to profile microbes from all three domains simultaneously. In order to obtain phylogenetic information for microorganisms across all three domains of life, separate target amplification and processing protocols are required [14], increasing the cost and analytical complexity of accurately assessing dynamic changes in the community across domains. Moreover, stochastic effects of primer interaction with a complex template, along with the difficulty in designing primers and amplification conditions that will equally target all members of a community [15], result in an unavoidable bias in community representation both in terms of presence/absence and relative abundances [15,16,17,18].

In recent years, metagenomic approaches in which whole nucleic acid recovered from a sample is fragmented and sequenced using “shotgun” methods have become increasingly popular. This approach has a significant advantage over methods based on amplification of taxonomic markers in that shotgun-sequencing data can overcome issues of bias and representation that are inherent in amplicon sequencing approaches, and it provides the additional advantage of describing the metabolic potential of the microbial community [19,20,21]. The sequencing of all DNA present in an environmental sample can, therefore, be considered a “gold standard” for taxonomic profiling. However, this approach is not without its own limitations. For example, it can be a wasteful enterprise in terms of the phylogenetic information recovered per sequencing cost. Shotgun sequencing is also not easily able to connect the functional potential observed in the sequencing data with the exact microbe within which that functionality resides. Additionally, DNA acquired from a community of microorganisms is inherently unbalanced; there are not equal numbers of each taxon, nor do all taxa have genomes that are of equal sizes. Thus, shotgun sequencing can provide a view of microbial community composition that is biased by genome size and microbial abundances. Overcoming this bias requires significant amounts of sequencing; therefore, chasing the rarity of the least abundant microbes by shotgun metagenomics sequencing carries a high financial cost [16,17,22,23]. The abundances of microbes within characterized complex microbial communities range over many orders of magnitude. While shotgun sequencing efforts provide a reasonable estimate of abundance, there is a significant loss in dynamic range when compared to PCR-based profiling.

The chaperonin 60 gene (*cpn60*) [10] (type I chaperonin) and its Archaeal homologue thermosome complex [24] (type II chaperonin) have been previously recognized as highly discriminating targets across all domains of life [25]. Pairwise identities of *cpn60* UT sequences accurately predict whole-genome sequence identities, and hence species affiliations [26]. Moreover, *cpn60* sequences have been used to discriminate subspecies of *Gardnerella vaginalis* [27], and have been shown to be a suitable taxonomic target for species and pathovar-level identification of plant pathogenic *Xanthomonas* spp. [28]. These sequences also meet standard International Barcode of Life criteria [29], and enable the *de novo* assembly of operational taxonomic units (OTU) from metagenomics sequence data [30]. OTU may be defined in various ways for different taxonomic targets, including clustering methods typically using 97% sequence identity as a cut-off [31], or methods based on sequence assembly that specify OTU as assembled sequences differing by a single nucleotide [29]. More recently, sub-OTU (sOTU) have been defined using Illumina sequence reads differing by a single nucleotide without assembly using read error correction (also called amplicon sequence variants, ASV) [32], which are suitable for both 16S [33] and *cpn60* [34] amplicon data. While “universal” PCR primers are available for *cpn60* [10,35], they are not expected to capture the pan-domain diversity of a complex microbial community through amplification. Moreover, *cpn60* amplification provides OTU abundances that do not always correlate to the true abundance of the microorganism in the sample [36]. If these limitations can be overcome, there is significant opportunity to dramatically improve research assessing host–microbiome interactions in plant, human and animal settings.

Recent advances in hybridization-based DNA capture combined with high throughput sequencing (CaptureSeq) have provided remarkably powerful means of enriching samples for DNA sequences of interest, including sequencing ancient pathogen genomes from the teeth of victims buried for centuries [37,38,39]. Moreover, Gasc and Peyret recently described a powerful method for isolating 16S rRNA-encoding gene sequences from environmental DNA samples using hybridization combined with Illumina sequencing [40]. These observations led us to consider the possibility of exploiting the unique features of *cpn60* to provide a microbial community profile without the use of universal PCR amplifications. A custom array of biotinylated RNA capture baits was designed based on the entire taxonomic composition of the chaperonin database cpnDB (www.cpndb.ca accessed on 1 April 2021) [10,41] and evaluated as a tool for enriching total genomic DNA simultaneously for type I and type II chaperonin target sequences. The taxonomic breadth of microbial species represented in cpnDB is nearly equivalent to 16S reference databases, although it does not contain as many entries [41]. The features of the CaptureSeq method were determined in relation to results obtained using shotgun metagenomic sequencing and the amplification of 16S rRNA-encoding genes on synthetic and natural microbial communities spanning a range of microbial ecosystems. Moreover, CaptureSeq was used to profile soil samples that have been treated with antibiotics over a 15-year time period. The results indicate that CaptureSeq provides the taxonomic reach associated with shotgun metagenomic sequencing combined with the sampling depth of amplicon-based sequencing.

## 2. Materials and Methods

### 2.1. CaptureSeq Array Design

Capture probes were designed based on all type I and type II chaperonin sequences in the public domain (i.e., cpnDB; www.cpndb.ca accessed on 1 April 2021; as of September 2013) [10]. In total, 15,733 probes were designed to be complementary to the type I and type II chaperone sequences. The design of probes was based on identifying 120 bp sequences from the reference database using a 60 bp incrementing step. Thus, the resulting probes share a 50% overlap with the next probe in a tiling-like fashion. The custom biotinylated RNA oligonucleotides were provided in equimolar concentrations as a pooled Mybaits array by Arbor Biosciences (Ann Arbor, MI, USA). Probe sequences are publicly available [42].

### 2.2. Template DNA Preparation

The Zymobiomics microbial community DNA standard consisting of 8 bacterial and 2 fungal genomic DNAs (cat. no. D6305) was obtained from Zymo Research (Irvine, CA, USA). This standard was diluted 1:20 as recommended by the manufacturer for 16S-based amplicon analysis and was also prepared for whole metagenome sequencing and CaptureSeq as described below.

A second synthetic microbial community was prepared using *cpn60* plasmids spiked into a naturally occurring microbial ecosystem. Background genomic DNA was prepared by washing wheat seeds and extracting DNA as previously described [25]. Amplicons corresponding to the *cpn60* universal target (*cpn60* UT) of 20 bacteria associated with the human vaginal tract [30] and known to be absent from the seed wash DNA background [25] were cloned into the pGEM-T Easy plasmid (Promega, Madison, WI, USA) and purified using the Qiagen Miniprep kit (Qiagen, Redwood City, CA, USA). The synthetic community was formed by combining equimolar concentrations of plasmids containing the *cpn60* UT for all 20 microorganisms [30]. Dilutions of this mixture (corresponding to 0.4, 0.04, and 0.004 ng plasmid DNA, or approximately 10^8^, 10^7^, and 10^6^ copies of each plasmid) were spiked into a background of 10 ng/µL of wheat seed carrier DNA. Spiked genomic DNA samples prepared in this way were sequenced using *cpn60* universal target amplification and CaptureSeq as described below.

Soil samples were obtained from a long-term study initiated in 1999 evaluating the effect of annual antibiotic exposure on soil microbial communities, described in [43]. Soil samples evaluated in the present study were obtained in 2013 following 15 sequential annual applications of a mixture of sulfamethazine, chlortetracycline and tylosin, each added at concentrations of 0.1, 1, or 10 mg kg^−1^ soil, along with untreated control plots. Soil was sampled 30 days after the spring application of antibiotics. The plots were planted with soybeans (*Glycine max*, v. Harosoy) immediately after incorporation of the antibiotics. Each treatment level was applied to triplicate plots yearly since 1999 as described [43]. Genomic DNA was extracted from 3.5 g of each soil sample using the PowerMax Soil DNA isolation kit (Mo-Bio Laboratories, Carlsbad, CA, USA) with a 5 mL elution volume. DNA extracts were quantified using a Qubit fluorimeter (Thermo Fisher Scientific, Waltham, MA, USA) and stored at −80 °C until processing and analysis.

A water sample was obtained from a pond located on a Saskatchewan farm (51.99° N, −106.46° W) on May 13, 2016. Biological material was recovered from 2 L of water by centrifugation at 20,000× *g* for 20 min. Total DNA was extracted from the pellet using a PowerWater DNA extraction kit (Mo-Bio Laboratories, Carlsbad, CA, USA) and quantified as described above.

Samples were obtained from bovine manure storage tanks after 28 weeks of storage, as part of a separate study examining the effects of storage parameters on the methanogenic communities. DNA was extracted from 1 mL of the slurry using a commercial kit (Qiagen).

### 2.3. Amplicon-Based Microbial Community Profiling

16S rRNA-encoding genes were amplified using primers 515f/926r under the recommended conditions, using the following cycling conditions: 1 × 94 °C, 3 min; 30 × 94 °C, 45 s, 50 °C 30 s, 72 °C 90 s; 1 × 72 °C 10 min [44]. The *cpn60* UT was amplified from DNA samples using 40 cycles of PCR with the type I chaperonin universal primer cocktail containing a 1:3 ratio of H279/H280:H1612/H1613 as described [35,45] and cycling conditions of 1 × 95 °C, 5 min; 40 × 95 °C 30 s, 42–60 °C 30 s, 72 °C 30 s; 1 × 72 °C 2 min. Replicate reactions from each amplification temperature for each sample were pooled and gel purified using the Blue Pippin Prep system (Sage Science, Beverly, MA, USA) with a 2% agarose cassette, and concentrated using Amicon 30K 0.5 mL spin columns (EMD Millipore, MA, USA). The 16S or *cpn60* UT amplicon from all samples was prepared for sequencing using the NEBNext Illumina library preparation kit (New England Biolabs), and sequenced using v2 Miseq chemistry.

### 2.4. Whole Metagenome and CaptureSeq Sample Preparation

Genomic DNA was diluted to 2.5 ng/µL and split into two aliquots of 100 µL each for shearing using a water bath sonicator as described [42]. Shotgun metagenomic sequencing libraries were prepared directly from one aliquot of each sheared genomic DNA sample using the NEBNext Ultra Illumina library preparation kit according to the manufacturer’s directions (New England Biolabs, Ipswich, MA, USA), which included limited PCR cycles (6–12) using adaptor-specific primers. Samples were then sequenced with 2 × 250 bp cycles of v2 Miseq chemistry (Illumina, San Diego, CA, USA). 

To generate the CaptureSeq libraries, the second aliquots of sheared genomic DNA samples were subjected to end repair and index addition using NEBNext as above, then hybridized to the capture probe array. A detailed protocol for CaptureSeq is provided elsewhere [42]. The chaperonin-enriched products were sequenced with 2 × 250 bp cycles of v2 Miseq chemistry (Illumina, CA, USA).

### 2.5. Reference Mapping

A reference database of all publicly available chaperonin sequences was generated by selecting a list of seven chaperonin protein sequences representing each taxonomic group: fungi, bacteria, archaea, plant mitochondria, plant chloroplast, and animal mitochondria. These sequences were used as queries for a BLAST search of GenBank using the default parameters to blastp. Matching protein sequences were manually vetted to generate a list of 30,141 protein identifiers. These protein identifiers were then used to retrieve the corresponding 30,120 nucleotide sequences available in GenBank according to the procedure described in Appendix A. The accession numbers of those nucleotide sequences are provided in Appendix A. The breadth of taxa that were retrieved by this method was similar to the taxonomic breadth represented in the 16S and ITS reference datasets (Appendix A). Sequencing reads from all samples were grouped into taxonomic clusters by paired local alignment to this reference set of chaperonin genes using bowtie2 (v. 2.2.3) [46]. The sequencing libraries were down-sampled to the size of the smallest shotgun metagenomic library (2777 mapped paired reads), and the number of reads mapping to each of the resulting taxonomic clusters was used as the basis for assessing the alpha and beta diversity metrics of the two profiling methods for equivalent sampling effort.

To compare the number of output sequencing reads for the different spiking levels, sequencing reads from the synthetic community-spiked samples were down-sampled to the smallest library size for each profiling technique (30,091 for amplicon and 506,247 for CaptureSeq) and mapped to a reference set of *cpn60* UT sequences for the 20 microorganisms in the panel by local paired alignment using bowtie2 as above.

### 2.6. Sequence Assembly

Read pairs from target taxonomic clusters obtained by reference mapping as described above were assembled de novo into *cpn60* OTU using Trinity (v. 2.4.0) with a kmer of 31, as described [30].

### 2.7. Sub-OTU (sOTU) Definition Using Amplicon Sequence Variants (ASV)

For 16S sequencing reads, primer sequences were trimmed using cutadapt (v2.8) [47], merged using FLASH2 (v2.00) [48], and ASV were determined using DADA2 [32]. For CaptureSeq reads, the *cpn60* sOTU was defined as nucleotides 1–220 of the *cpn60* UT (after trimming the 5′ amplification primer). Primer sequences were removed and all reads were trimmed to 220 bp using cutadapt. Sequences in the reverse orientation were reverse complemented prior to ASV analysis with DADA2.

### 2.8. Alpha Diversity Analysis

To compare the richness and diversity metrics between the three profiling techniques, mapped sequencing reads were down-sampled from 250–2750 reads to simulate a uniform sampling effort across profiling techniques. Metrics were averaged across 100 bootstrapped datasets using the multiple_rarefactions.py and alpha_diversity.py scripts from QIIME (v. 1.8.0) [49]. Statistical significance between alpha diversity metrics was determined using the Kruskal–Wallis rank-sum test and equality of variances was evaluated using Lavene’s test from the stats and rstatix packages in R.

In the cases where the total effect of sequencing effort was required for comparisons across estimates of community coverage, read thresholds were transformed to reflect total sequencing effort for each sample. 

### 2.9. Beta Diversity Analysis

To compare the community similarity between different sequencing methods, mapped sequencing reads were down-sampled to the size of the smallest metagenomic library sample (2777 mapped reads). For intra-technique comparisons, mapped sequencing reads were down-sampled to the smallest library size within each profiling method; 2777 for metagenomic, 127,642 for CaptureSeq, and 27,388 reads for amplicon libraries. Principal Coordinate Analysis of inter- and intra-technique Bray–Curtis distance was calculated using the vegan package (v. 2.4.2) in R (v. 3.2.4).

### 2.10. OTU Quantification

Assembled OTU-specific primer and hydrolysis probe sets were designed using Primer3 [50] or Beacon Designer (v.7) (Premier Biosoft, Palo Alto, CA, USA) as described previously [51]. Annealing temperatures were optimized for each reaction using gradient PCR with ddPCR Supermix for Probes (Bio-Rad, Mississauga, ON, Canada) using 900 nM each primer and 250 nM of hydrolysis probe in a 20 µL reaction volume. Primer/probe sequences and optimized amplification conditions are shown in Appendix A. Template DNA was digested prior to amplification using *Eco*RI at 37 °C for 60 min. A final volume of 2–5 µL was used as template for droplet digital PCR (ddPCR). Emulsions were formed using a QX100 droplet generator (Bio-Rad, Hercules, CA, USA), and amplifications were carried out using a C1000 Touch thermocyler (Bio-Rad). Reactions were analyzed using a QX100 droplet reader (Bio-Rad) and quantified using QuantaSoft (v.1.6.6) (Bio-Rad). Results were converted to copy number/g soil extracted by accounting for sample preparation and dilution. For the prepared CaptureSeq libraries, results were converted to copy number/µL by considering dilution factors.

OTU corresponding to the *cpn60* UT plasmids added to the wheat seed wash background were quantified using real-time quantitative PCR (qPCR) primers and amplification conditions, as described previously [52]. Total bacteria were enumerated using qPCR targeting the 16S ribsosomal RNA-encoding gene as described previously [53].

## 3. Results

### 3.1. CaptureSeq Provides Microbial Community Profiles from Synthetic Microbial Ecosystems

#### 3.1.1. Zymobiomics Reference Panel

A simulated microbial community consisting of genomic DNA from eight bacteria and two eukaryotes (with a theoretical composition of 12% each of the eight bacterial genomes, and 2% each of the two eukaryotic genomes) was examined using CaptureSeq, 16S rRNA-encoding gene amplification, and whole metagenome sequencing. To facilitate a direct comparison between 16S amplicon sequencing and CaptureSeq, results were generated using ASV analysis for both methods. Considering only the bacterial genomes that are accessible with 16S amplicon analysis, both methods successfully detected all eight bacterial OTU (Figure 1). While the means of the observed proportional composition of the artificial community were identical between the two methods (12.5% across the eight genomes), CaptureSeq provided a more reproducible profile, with less variation from the mean observed (Figure 1). In general, CaptureSeq provided more data from the higher G/C content bacteria compared to 16S amplification, with the highest G/C bacterium (*P. aeruginosa*, 66.2% G/C) found in nearly double the proportional abundance by CaptureSeq compared to 16S amplification. Conversely, *S. aureus* (32.7% G/C) was more accurately represented by 16S amplification compared to CaptureSeq. Overall, however, both methods provided complete and accurate coverage of the bacterial species present in the artificial community.

Considering the complete synthetic microbial community, while 16S was unable to identify either of the yeasts present in the mixture, CaptureSeq data examined using ASV analysis identified one of the two yeasts, with ASVs corresponding to *C. neoformans* represented in the CaptureSeq data (Table 1). These ASVs included the beginning of a 64 bp intron found between nucleotides 165 and 166 of the 555 bp *C. neoformans cpn60* universal target (UT) sequence (cpnDB ID b5732). However, no ASVs corresponding to *S. cerevisiae cpn60* were found in the CaptureSeq data.

In addition to ASV analysis, the CaptureSeq data were also used to identify OTU in the artificial community using a reference mapping and de novo assembly approach. Using this method, all eight bacterial *cpn60* sequences and both of the yeast *cpn60* sequences were identified in the CaptureSeq data (Table 1). In every case, the assembled OTU length considerably exceeded the length of the *cpn60* UT that is accessed using the *cpn60* universal PCR primers, and, therefore, identified the complete UT sequence plus flanking sequences of varying length.

The artificial community was also analyzed using whole metagenome sequencing. This approach identified all of the microorganisms represented in the synthetic microbial community, including all eight bacteria and both fungi (Table 1). These results were nearly identical to those observed using *cpn60*-based CaptureSeq, except that the assembled fragments tended to be longer and more accurately assembled using CaptureSeq compared to shotgun metagenomic sequencing (Table 1).

#### 3.1.2. Quantification of Microbial Abundances in CaptureSeq

Using a second synthetic microbial community consisting of 20 *cpn60* UT plasmids spiked into background DNA from a cereal seed wash facilitated a quantitative examination of microbial community profiles generated by CaptureSeq. qPCR quantification of *cpn60* targets from the synthetic community before and after hybridization revealed an enrichment of 3–4 orders of magnitude for *cpn60*-containing DNA fragments compared to 16S rRNA-encoding genes (Table 2). For the five exogenous microorganisms that were quantified, the observed ratio of cpn60/16S reads increased consistently with hybridization at all spike levels (Table 2). Endogenous microorganisms representing a bacterium (*P. agglomerans*) and a fungus (*Alternaria* sp.) were also detected that corresponded to those previously identified in wheat seed wash samples [25], and these also showed increased cpn60/16S ratios after hybridization (Table 2). Sequencing of the post-hybridization samples provided a measurement of the number of reads generated for each of the five bacterial *cpn60* plasmids at each spike level. For each of the five targets, sequencing reads provided by CaptureSeq data correlated strongly to the qPCR-determined abundances, with significant Pearson correlation coefficients (r^2^) ranging from 0.998–1.000 (Table 3). Similarly, PCR amplification of *cpn60* generated sequencing read numbers that correlated with qPCR results, but with lower coefficients that were not significant (Table 3). Across the 5 targets at all levels, both methods generated statistically significant Spearman correlations between qPCR-determined abundances and sequencing read counts (Table 3).

CaptureSeq generated profiles that accurately reflected the relative amounts of DNA spiked into the seed wash background (Appendix A). In the CaptureSeq libraries, the number of mapped sequencing reads for each member of synthetic community was within one order of magnitude from the mean for each spike level (Appendix A). *De novo* assembly of the mapped sequencing reads for each microorganism from the *cpn60* plasmid synthetic panel generated OTU that were >99% identical to the known *cpn60* sequences (not shown). Based on the results observed using these synthetic microbial communities, analysis of natural microbial communities used the reference mapping and de novo assembly approach. Microbial communities from natural microbial ecosystems were profiled using *cpn60*-based CaptureSeq, *cpn60* universal target amplification, and whole metagenome sequencing.

### 3.2. CaptureSeq Provides Microbial Community Profiles in Natural Ecosystems

Microbial profiles were generated using all three methods from natural environmental ecosystems including soil, manure storage tanks, and a non-aerated terrestrial pond. These samples were chosen to reflect high complexity, and principally Bacteria (soil), samples enriched in Archaea (manure storage tank), and samples with higher numbers of Eukarya (freshwater pond). The CaptureSeq profiles of these communities provided a taxonomic overview of Bacteria, Archaea and Eukarya simultaneously, and contained sequencing reads from 9361 (soil), 9306 (manure), and 6568 (pond) distinct taxonomic clusters (Appendix A). Additionally, the CaptureSeq profiles facilitated inter-domain comparisons of read abundances among taxonomic groups, since the abundances could be expressed in relation to the total pan-domain community as opposed to reflecting only the proportions within a single domain (Figure 2). 

The soil sample microbiomes, as expected, were composed primarily of Bacteria, with Proteobacteria and Actinobacteria comprising 60% and 25% of the pan-domain community, respectively. Consistent with observations in the synthetic communities, high G/C Actinobacteria were well represented in the microbial community profiles generated by CaptureSeq for the soil samples, including several members of the genera *Nocardiodes*, *Marmoricola*, and *Pseudonocardia* with G/C contents ranging from 64–71% (not shown). Members of the phyla Acidobacteria and Gemmatinomonadetes represented an additional 5% each of the soil microbiome. Total archaeal reads only accounted for 0.03–0.08% of the soil microbial community; however, there were still 165 archaeal taxonomic clusters identified in the soil. Eukarya represented just 0.18–0.21% of the soil microbiome, with Fungi and Metazoa being the most abundant taxonomic groups. While the bovine manure storage tank-derived samples also contained a diverse array of Bacteria, they only represented 77–80% of the microbiome, compared to > 99% for the soil samples. CaptureSeq libraries from the manure storage tank samples contained 19–22% archaeal reads, of which the vast majority were methanogens from the Phylum Euryarchaeota. The terrestrial pond contained a much greater proportion and diversity of Eukaryotes, representing 6.7% of the sequencing reads and 361 taxonomic clusters (Appendix A).

### 3.3. CaptureSeq Data Facilitates the Assembly of Target OTU from Taxonomic Clusters

Analysis of the synthetic microbial communities demonstrated that the *cpn60* molecular barcode could be reconstituted from CaptureSeq data using the de novo assembly of OTU. To determine if the assembly of OTU representing individual organisms was reliable in complex natural microbial communities using CaptureSeq, we selected target taxonomic clusters identified through reference mapping for assembly. The *de novo* assembly of eukaryotic sequencing reads from the terrestrial pond sample generated 11 out, most closely related to members of the Phylum Chlorophyta (green algae). Additionally, the assembly of OTU most similar to *Anopheles* sp. (mosquitoes), and three members of the Phylum Alveolata (protists), suggested that CaptureSeq was able to retrieve *cpn60* DNA from a diverse array of Eukarya. Compared to reference sequences in cpnDB, these de novo assembled OTU had nucleotide identities ranging from 59–84%, suggesting that the current probe array design and hybridization conditions were sufficiently permissive to allow for the capture of novel *cpn60* sequences (true unknowns).

To examine the suitability of *de novo* assembly for providing taxonomic markers suitable for tracking the abundances of particular OTU across samples, we selected target microorganisms for quantification in antibiotic-treated soil samples using OTU-specific qPCR. For Bacteria, we quantified the *Microbacterium lacus* strain C448, which was previously cultured from these soil samples and shown to degrade and metabolize the sulfonamide antibiotic added to the field plots [54]. While the presence of this target in the soil samples was confirmed using culture methods, it was under-represented in the shotgun metagenomic libraries when compared to the CaptureSeq profiles. Only the CaptureSeq data provided a sufficient number of target sequencing reads for de novo assembly, generating a 1066 bp OTU that was >99% identical to the *cpn60* sequence obtained from the genome of this organism [55]. Quantification of *M. lacus* C448 showed that the bacterium was present at a low level in all soil samples of between 10^3^ and 10^4^ gene copies per gram of soil, and that the levels were significantly higher in the 10 ppm antibiotic-treated soil samples compared to untreated soils (Table 4).

A *cpn60* OTU corresponding to *Acinetobacter baumanii/A. calcoaceticus* was also assembled from the CaptureSeq data and quantified using ddPCR. This OTU was determined by CaptureSeq to be maximally abundant in plot 4, which was 1 of 3 replicates that had been treated with 10 ppm of antibiotics. Consistent with the CaptureSeq data, ddPCR revealed that the OTU corresponding to *A. baumanii* had by far the highest abundance in plot 4 (approximately 10^6^ genomes/g soil) and was present at levels near or below 10^3^ genomes/g soil in other soil samples in which reads corresponding to this OTU were not detected (Figure 3). Plots 11 (10 ppm) and 3 (0.1 ppm) also had *A. baumanii* OTU in the CaptureSeq datasets and this OTU was present at slightly higher levels in these plots (Figure 3). 

Assembled OTU targets from the domains Eukarya (type I-*Phythophthora infestans*) and Archaea (type II-*Methanoculleus* sp.) were also selected for quantification using ddPCR. The archaeal OTU was quantified at levels between 495 and 527 gene copies per gram of soil. The OTU corresponding to *P. infestans* was also present at very low levels in these samples, yet was detectable by CaptureSeq (Table 4). These results suggest that the CaptureSeq method could sample complex microbial communities with a limit of detection within the dynamic range of even very sensitive quantification methods such as ddPCR.

### 3.4. CaptureSeq Provides Sequencing Depth That Is Similar to That Achieved by Taxonomic Marker PCR Amplification

The number of sequencing reads corresponding to *cpn60* genes represented 0.07% of the total reads from the whole metagenome library compared to an average of 16.7% (±0.8%) for CaptureSeq and 94.8% (±0.6%) for amplicon libraries (Appendix A). Examining the Good’s coverage estimator [56] as a function of sequencing depth revealed that CaptureSeq and amplicon-based profiling provided almost complete community coverage in soil, while whole metagenome sequencing required much greater sequencing effort to achieve a high level of community coverage (Figure 4). These observations led us to compare the relative cost estimates among these methods, which showed that CaptureSeq carries a cost that is between that of *cpn60*-based amplicon profiling and shotgun metageomics (Appendix A).

### 3.5. Microbial Ecosystem Diversity Metrics of Antibiotic-Treated Soil Samples

To compare the microbial community diversity metrics obtained using the *cpn60* amplicon, whole metagenome sequencing, and CaptureSeq methods, we focussed our analyses on the untreated soil samples and the samples treated with the highest concentration of antibiotic (10 ppm). The objective of this analysis was not to obtain an accurate measurement of the alpha diversity in the soil samples, but to compare both the calculated values and the variability among biological replicates observed using the different profiling methods at equivalent sampling depths. Therefore, we determined these values only up to the number of *cpn60* reads that were observed using shotgun community sequencing. Soil community richness (Chao1) (*p* = 0.004), evenness (Simpson 1-D) (*p* = 0.004) and diversity (Shannon H’) (*p* = 0.004) metrics were all significantly higher when profiled using whole metagenome sequencing compared to CaptureSeq (Figure 5). Additionally, the Simpson and Chao1 alpha diversity metrics of the CaptureSeq method showed the lowest variance among the biological replicates of each treatment, even when libraries were down-sampled to very low levels (Figure 5). CaptureSeq provided Shannon and Simpson indices that were between those determined using shotgun metagenomics and amplicon sequencing (Figure 5). The alpha diversity metrics were not significantly different between the antibiotic treated and untreated soil samples using any of the analysis methods, but the low sampling levels used for comparison among methods precluded the accurate measurement of these parameters. 

Hierarchical clustering of OTU abundance patterns using data obtained from each of the three methods revealed that the CaptureSeq and shotgun metagenomic datasets displayed patterns of microbial abundances that were more similar to one another and that both were distinct from the pattern shown by the *cpn60* amplicon datasets (Figure 6). Moreover, of the three methods analyzed, only CaptureSeq showed a hierarchical clustering pattern that suggested a possible treatment effect of the 10 ppm antibiotic compared to untreated soil samples (Figure 6). Similarly, when intra-technique beta diversity was assessed using principal coordinate analysis, only the CaptureSeq data provided measures that suggested a possible treatment effect of the 10 ppm antibiotic in the long-term treated soil samples (Figure 7).

## 4. Discussion

Experimental approaches to determining the taxonomic composition of microbial ecosystems have typically employed PCR amplification of Domain-restricted taxonomic markers, typically various regions of the 16S rRNA-encoding gene for Bacteria [44] and the ITS locus for Fungi [13]. In addition, PCR amplification of the *cpn60* UT can generate microbial community profiles that include bacteria and eukaryotes simultaneously [25]. While these methods can provide a rapid and cost-effective means of determining the taxonomic composition of a microbial community, they also have well recognized limitations associated with low taxonomic resolution and PCR amplification biases [33,57]. These drawbacks can be partially overcome with whole metagenome shotgun sequencing [58], but this approach is not feasible in many experimental models. These considerations have led to the development of several alternative means of profiling microbial communities, such as 16S rRNA-encoding gene-based hybridization capture [40], transfer RNA sequencing [59], and real-time single molecule sequencing of full-length 16S rRNA-encoding genes [9]. Each of these methods presents unique advantages aimed at overcoming some of the limitations of PCR-based profiling, but each method also comes with its own limitations that must be considered when deciding on an appropriate method for a particular experimental system.

Here, we have described an experimental approach for determining the taxonomic composition of microbial communities that exploits the features of the solution-based hybridization of *cpn60* sequences. This approach utilizes the advantages of *cpn60*-based microbial community profiling, including its high taxonomic resolution and ability to profile multiple domains simultaneously [25,60,61,62]. While *cpn60* has proven to be a powerful and useful taxonomic marker for microbial community analysis, PCR amplification of *cpn60* targets in microbial communities presents analogous biases in community representation, as seen in 16S and ITS-based amplification strategies [63]. 

This work provides experimental and computational methods for analyzing microbial community composition using *cpn60*-based hybridization and demonstrates that CaptureSeq provides quantitatively reliable data that can detect OTU from multiple domains within the same dataset. Nevertheless, improvements to both the experimental and computational parts of the workflow can be considered. On the experimental side, the composition of the hybridization array could be optimized and expanded based on the accumulation of novel *cpn60* sequences from currently unrepresented taxa. This would help to minimize biases associated with the lack of representation of taxa with no probes on the capture array of sufficient nucleotide sequence similarity. The hybridization conditions could also potentially be optimized to maximize the recovery of *cpn60* sequences in the dataset. The current workflow resulted in an approximately 200-fold enrichment of the soil samples for the taxonomic marker of interest, from under 0.1% of reads in the shotgun metagenomic sequencing to over 15% of reads in the CaptureSeq datasets. While this level of enrichment enabled deep sampling of the soil microbial communities (similar to that attained using PCR-based enrichment), it does not approach the levels of enrichment that were observed using 16S-based hybridization [40]. More recent CaptureSeq experiments employing newer blocking oligonucleotides provided by the manufacturer has resulted in improvements to the recovery of *cpn60* gene fragments–up to 70% of reads in recent studies (data not shown). 

Computationally, a key challenge to be overcome in the analysis of CaptureSeq data, such as PCR amplification-based methods, is the most appropriate means of defining an OTU. We have demonstrated here that the assembly of *cpn60* gene fragments exceeding even the length of the 549–567 bp *cpn60* UT [41] is possible using CaptureSeq data, but assembly algorithms can be computationally challenging on large datasets, and the formation of artefactual OTU can be difficult to avoid. Targeted assembly of specific OTU from reference bins obtained by mapping offers a means to overcome these difficulties and can provide taxonomic markers suitable for strain tracking across experimental systems and for strain culture. Moreover, other assembly methods are available that do not require prior reference binning; for example, we have successfully assembled *cpn60* OTU from complex CaptureSeq data without reference mapping using TransAbyss [64]. Nevertheless, alternative means of defining OTU have been described for 16S PCR amplicon data that exploit unique sequence variants and are recommended for nearly all applications [33]. CaptureSeq can be used to define analogous *cpn60*-based sub-OTU (sOTU) using DADA2 [32]; however, the read length of Illumina data precludes the use of the entire ~550 bp *cpn60* UT sequence for OTU definition. Recent work has determined that the first 150 bp of the *cpn60* UT is suitable for sOTU definition [34]. However, the ASV method appeared to have a lower sensitivity compared to assembly, as it failed to detect one of the eukaryotic OTU from the Zymobiomics reference panel. Further refinement of sOTU definition is a requirement for future CaptureSeq studies. We suggest that the sOTU approach may be suitable for experimental questions aimed at determining and comparing total microbial community structure across samples, while the assembly approach may be more suitable for experimental questions aimed at identifying particular microorganisms that may be associated with some desired function—this is because the assembly approach provides more taxonomic information for microorganism identification (by culture) and tracking across experimental systems due to the length of the assembled *cpn60* OTU. The use of full-length *cpn60* UT sequences will provide more precise species-level assignments and will allow us to build a reference database of experimentally observed *cpn60* sequences for comparisons across environments. 

The effects of long-term antibiotic treatments on soil microbial communities was not a primary focus of this study. Nevertheless, using CaptureSeq, we observed broad differences in the taxonomic composition and beta diversity of microbial communities in soil samples treated with antibiotics compared to untreated soil, with antibiotic-treated communities clustering distinctly from the untreated soil samples. The strong correlations we observed between OTU abundances and qPCR-determined abundances in CaptureSeq data in the synthetic panel, along with the agreement between CaptureSeq read frequencies and ddPCR-determined abundances for the bacteria analyzed in the natural soil ecosystem (*Microbacterium* and *Acinetobacter*), suggest that the CaptureSeq data accurately represented the composition of these microbial communities. The fact that known antibiotic-degrading bacteria (*M. lacus*) were detected in higher abundances in the antibiotic treated soil is consistent with the selection of such bacteria in the presence of antibiotics. In addition, the increased abundance of *A. baumanii* in antibiotic treated soils, which was noted by CaptureSeq and by ddPCR, is consistent with the known ability of this organism to acquire antibiotic resistance genes through horizontal gene transfer [65]. 

While the experimental and analytical methods for CaptureSeq could be improved, the method offers several distinct advantages compared to PCR amplification and shotgun metagenomics for analyzing microbial communities. For example, CaptureSeq provided a balanced view of the relative abundances of microorganisms within the community. PCR-associated representational bias, which presents a skewed representation of microbial taxon abundance [66], is a well-known phenomenon [67,68,69]. CaptureSeq also resulted in an improvement in the representation of high G/C content microorganisms compared to amplification. Difficulty in the amplification of high G/C content targets is a phenomenon that has been previously observed using both 16S and *cpn60* taxonomic markers from mixed communities [35,70]. The de novo assembly of taxonomic clusters from the CaptureSeq datasets into OTU for which probes were not explicitly designed, such as *M. lacus* strain C448, also suggests that off-target *cpn60* sequence capture can expand the breadth of OTU observed in the dataset beyond the sequences represented in the probe array and can include sequences that have not been previously observed. Off-target hybridization resulting in the identification of novel taxa was also observed using 16S rRNA gene-based hybridization [40].

Both CaptureSeq and whole metagenome sequencing provided the means to identify OTU from all domains simultaneously. The ability to calculate the abundances of organisms as a proportion of the entire pan-domain community facilitates the identification of inter-domain relationships and syntrophies. This is of particular importance in many settings (e.g., manure or gut health) in identifying the syntrophic relationships between volatile fatty acid producing Bacteria and methanogenic Archaea [71]. In soil, the complex relationship between saprophytic Fungi and Bacteria is critical to examining the role of the microbiome in nutrient cycling [72]. This advantage is not offered using amplification of universal targets, although PCR-based enrichment does provide the benefit of very deep coverage of complex microbial communities. Whole metagenome sequencing does not provide the community coverage of the CaptureSeq method at a similar sequencing effort, suggesting that complex microbiomes will likely require additional phylogenetic data to make any informed examination of microbial diversity metrics. Whole metagenome sequencing can reasonably be considered to be a less biased means of determining the taxonomic composition of an environmental sample, and may be a suitable choice when sufficient sequencing resources are available. However, the abiding popularity of amplicon-based profiling is at least partially a result of the high degree of enrichment of taxonomically informative sequence reads that it generates. CaptureSeq provides an alternative that avoids amplification biases associated with PCR while retaining the sequencing efficiency of amplicon-based profiling. 

Molecular microbial community profiling is one of the foundational steps in exploring microbiome structure–function relationships in an experimental system [73,74,75]. To generate and evaluate scientific hypotheses, it is critical to generate a microbiome profile that reflects the natural state as closely as possible with sufficient sensitivity to evaluate both abundant and rare microorganisms. The *cpn60*-based method described herein permits taxonomically broad and deep microbial community profiling of complex microbiomes. Thus, CaptureSeq has the potential to impact life sciences research wherever microbes are thought to be important, including human health and nutrition [76], agriculture [77], biotechnology [78], and environmental sciences [79]. In each of these areas, researchers can choose from an increasing array of tools to address the particular experimental question at hand. While all microbial community profiling techniques have inherent limitations and biases, CaptureSeq is a suitable alternative that provides quantitative, cross domain data for the analysis of complex microbial ecosystems.

## 5. Conclusions

In this work, we have demonstrated the utility of *cpn60*-based hybridization to enrich environmental DNA samples for taxonomically informative DNA sequences. Using synthetic microbial ecosystems, CaptureSeq was shown to provide robust results regarding the presence and abundance of prokaryotes and eukaryotes simultaneously, and the read abundances generated were quantitatively reliable. CaptureSeq was also shown to provide pan-domain microbial community profiles in complex natural ecosystems and generate read abundances that were consistent with observations made using microorganism-targeted molecular diagnostic assays. While both the experimental and computational methods could be improved, this work represents a first step in demonstrating the utility of this approach for providing microbial community profiles that reflect the natural state more closely than PCR-based methods. CaptureSeq could easily be applied to a wide range of microbial ecosystems, providing robust data on microorganism presence and abundance that can inform many experimental questions regarding the role of the microbiota in human, environmental, and agricultural ecosystems.

## Figures and Tables

**Figure 1 microorganisms-09-00816-f001:**
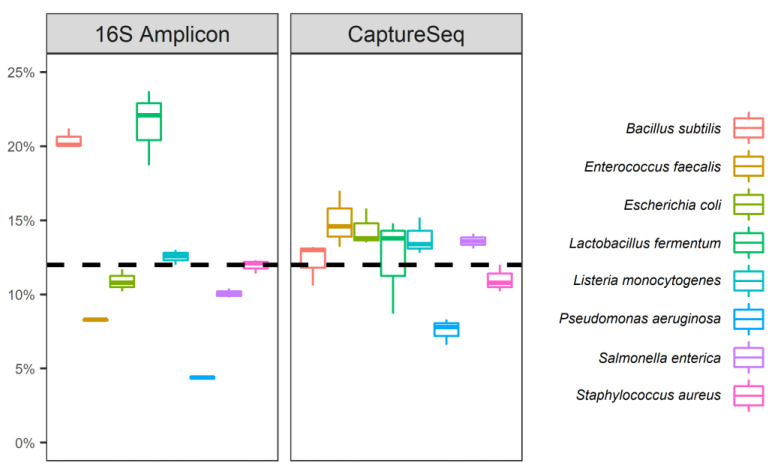
Examination of an artificial microbial community (Zymobiomics Microbial Community Standard) using 16S rRNA-encoding gene amplification and CaptureSeq. The relative abundances of each of the 8 bacterial OTU present in the synthetic community are shown for each of 3 replicates of each method compared to the theoretical composition provided by the manufacturer (dashed line).

**Figure 2 microorganisms-09-00816-f002:**
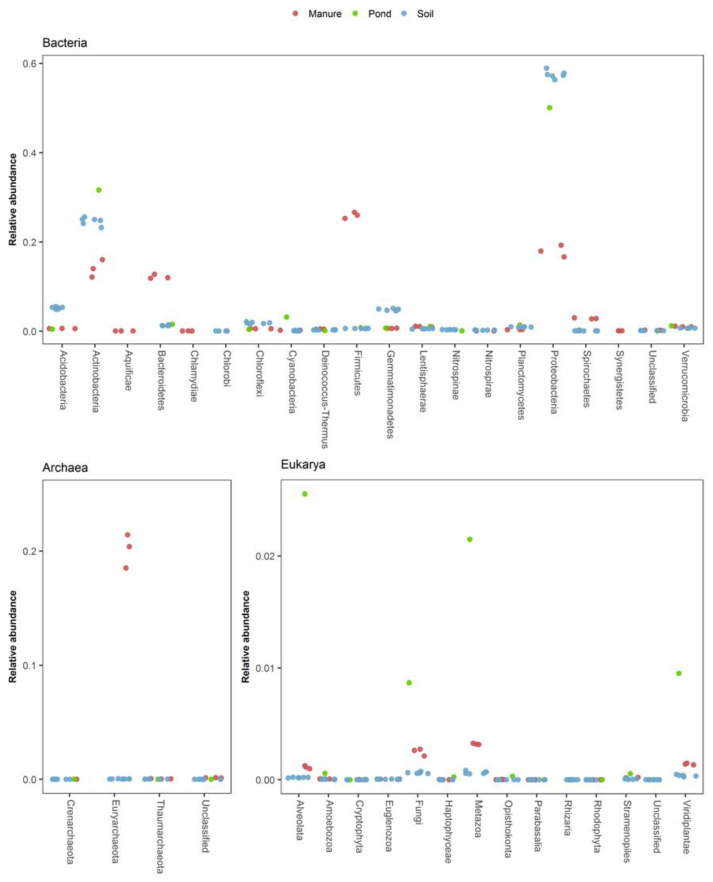
CaptureSeq results obtained by reference mapping on an ecologically diverse range of samples including soil (*n* = 6); manure storage tanks (*n* = 3); and a freshwater pond (*n* = 1). The relative abundances of individual phyla are expressed as a proportion of the entire pan-domain microbial communities.

**Figure 3 microorganisms-09-00816-f003:**
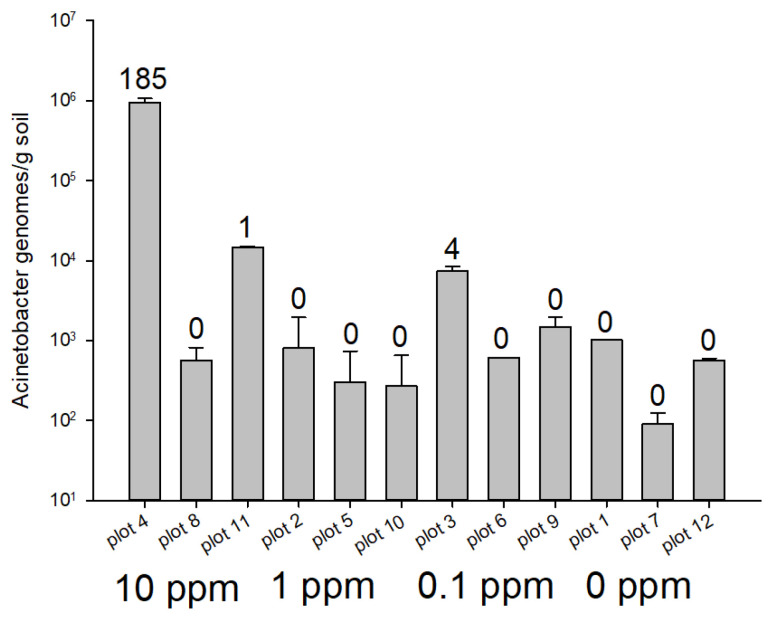
Quantification of an assembled OTU corresponding to *Acinetobacter baumanii/A. calcoaceticus* in soil samples treated with antibiotics. The number of CaptureSeq reads mapping to this taxonomic cluster in each soil sample is indicated above each bar.

**Figure 4 microorganisms-09-00816-f004:**
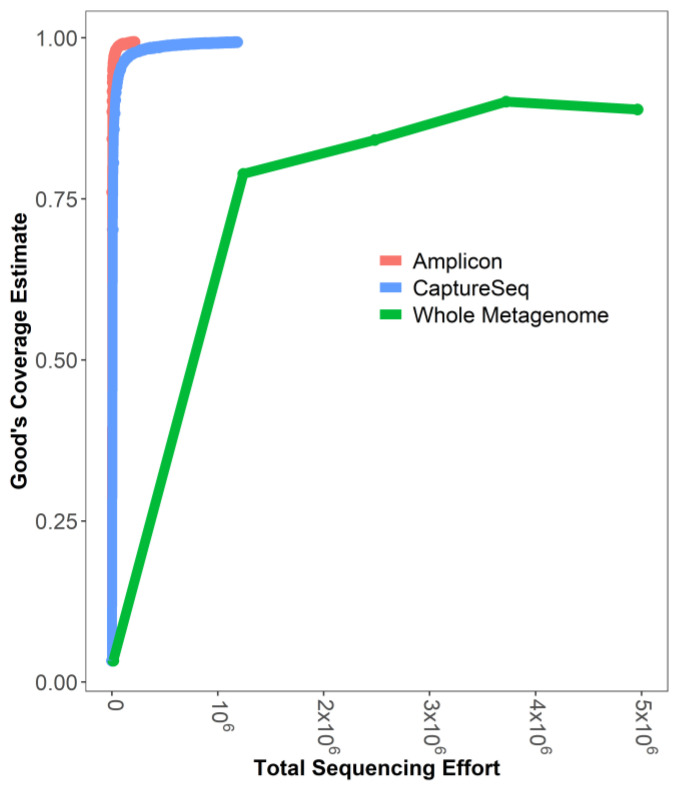
Good’s coverage estimate as a function of the average total sequencing effort for six soil samples each profiled using amplicon (red), CaptureSeq (blue), or shotgun metagenomic (green) approaches.

**Figure 5 microorganisms-09-00816-f005:**
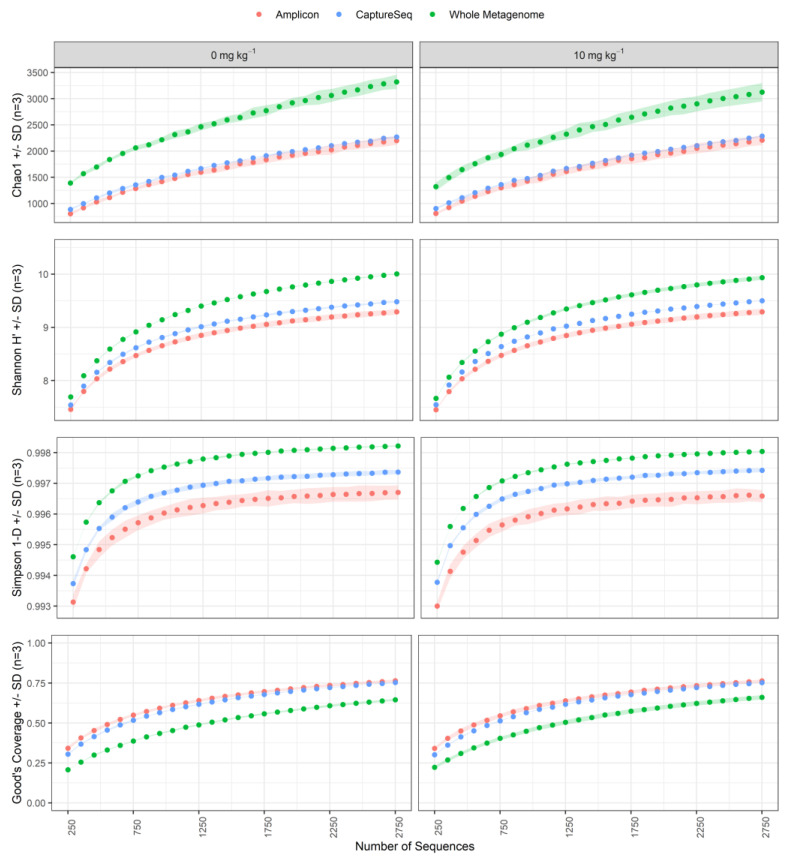
Alpha diversity metrics for soil samples profiled using *cpn60* amplicon (red), CaptureSeq (blue), or shotgun metagenomic (green) approaches. Metrics were calculated using libraries that were down-sampled from 250–2750 reads and were averaged across 100 bootstrapped datasets. The shaded area corresponds to the standard deviation of the three replicate soil plots for each antibiotic treatment condition.

**Figure 6 microorganisms-09-00816-f006:**
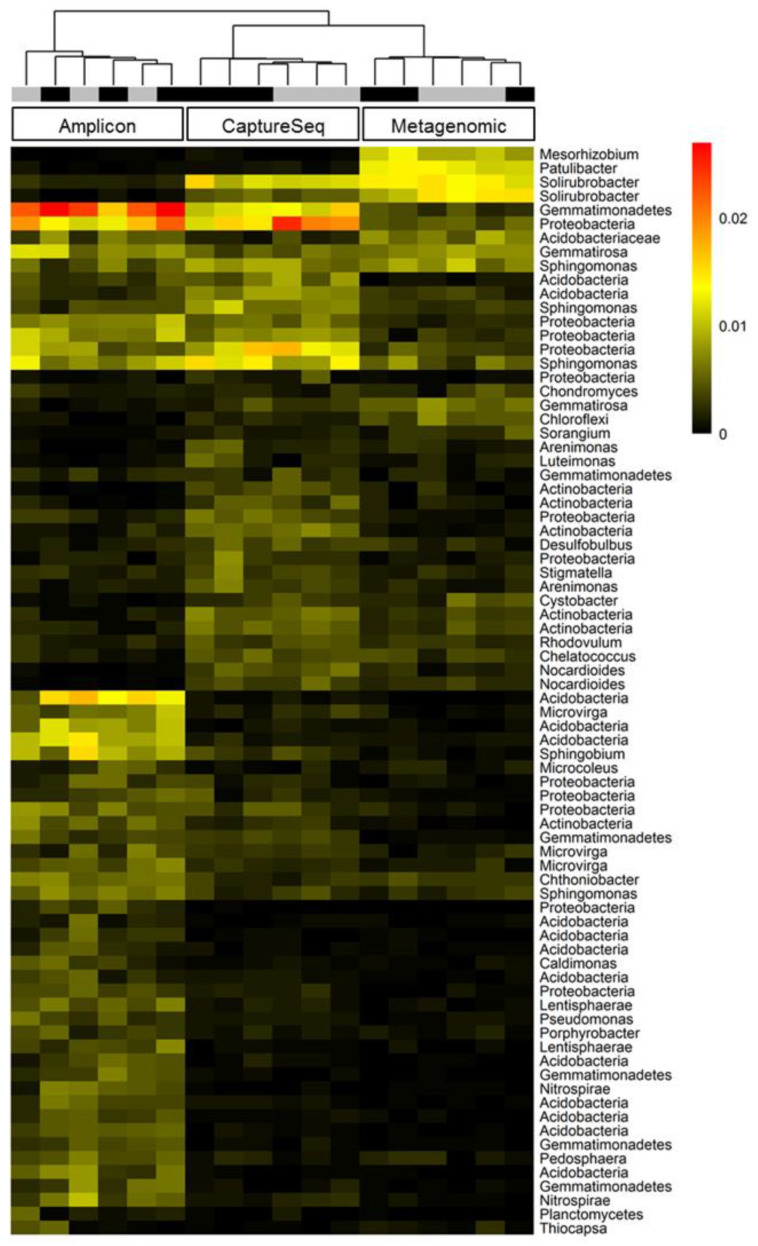
Proportional abundances of taxonomic clusters obtained by reference mapping for type I chaperonins in soil samples profiled using amplicon, CaptureSeq, or shotgun metagenomic approaches. Samples were clustered based on Bray–Curtis distance, and reference clusters composing a minimum of 0.5% of the mapped sequencing reads in any one sample are shown. Samples are coded according to antibiotic treatment: black (10 ppm) or gray (0 ppm).

**Figure 7 microorganisms-09-00816-f007:**
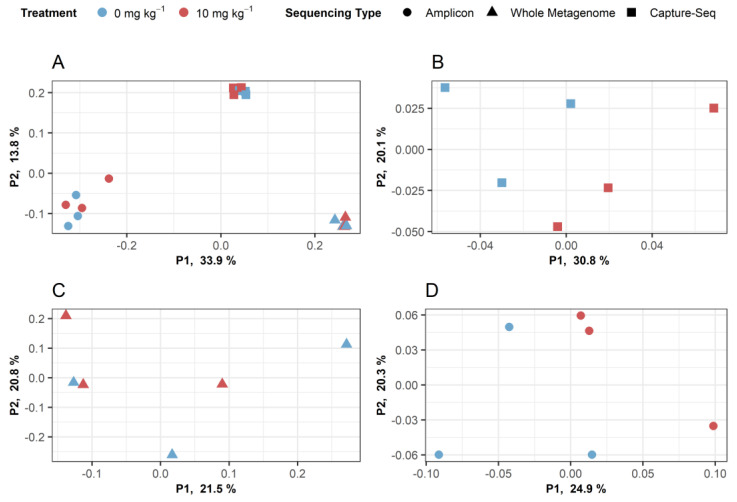
Principal coordinate analysis of Bray–Curtis dissimilarity between soil samples profiled using CaptureSeq (square), shotgun metagenomics (triangle), or *cpn60* amplicon (circle) approaches. OTU frequencies were determined by reference mapping as described in the text. (**A**) All approaches considered together. (**B**) CaptureSeq. (**C**) Shotgun metagenomics. (**D**) *cpn60* amplicon.

**Table 1 microorganisms-09-00816-t001:** Analysis of Zymobiomics microbial community DNA standard using *cpn60*-based CaptureSeq, shotgun sequencing, and taxonomic marker gene amplification.

		CaptureSeq Assembly ^1^			Shotgun Assembly ^2^			sOTU Detected ^3^	
Organism	*cpn60* UT Sequence Length	OTU Detected	Assembly Length (bp)	Sequence Identity ^4^	*cpn60* OTU Detected	Assembly Length (bp)	Sequence Identity	*cpn60* CaptureSeq	16S Amplification
Prokaryotes									
*Bacillus subtilis*	552	+	2144	100%	+	873	100%	+	+
*Escherichia coli*	555	+	1138	97%	+	1208	99%	+	+
*Enterococcus faecalis*	552	+	1235	100%	+	1133	100%	+	+
*Lactobacillus fermentum*	552	+	2165	100%	+	870	100%	+	+
*Listeria monocytogenes*	552	+	1589	100%	+	811	99%	+	**+**
*Pseudomonas aeruginosa*	555	+	1183	100%	+	668	93%	+	+
*Staphylococcus aureus*	552	+	1849	100%	+	487	99%	+	+
*Salmonella enterica*	555	+	1138	97%	+	1366	100%	+	**+**
Eukaryotes									
*Saccharomyces cerevisiae*	555	+	1301	100%	+	384	94%	NF ^5^	NF
*Cryptococcus neoformans*	619 ^6^	+	1435	100%	+	549	94%	+	NF

^1^ OTU defined by assembly of CaptureSeq reads using Trinity as described in Methods. ^2^ OTU defined by assembly of shotgun metagenomic reads using Trinity as described in Methods. ^3^ sOTU, sub-OTU as defined by amplicon sequence variant (ASV) analysis using deblur/DADA2 as described in Methods. ^4^ Sequence identity between the assembled OTU and the *cpn60* sequence of the corresponding species. Contains *cpn60* UT and flanking sequences. ^5^ NF, not found. ^6^
*cpn60* UT of *C. neoformans* contains a 64 bp intron, which was found in the assembled OTU and in the ASV sequence.

**Table 2 microorganisms-09-00816-t002:** qPCR-determined log_10_
*cpn60* gene copies in wheat seed wash samples spiked with varying amounts of *cpn60* plasmids from non-endogenous microorganisms. Two microorganisms previously shown to be associated with the wheat seed microbiota, along with the total number of 16S gene copies (representing non-*cpn60* DNA) are also shown. Results are shown prior to hybridization (pre-hyb) and after hybridization (post-hyb). The observed ratio of *cpn60*/16S genes is shown in brackets for each target.

Spike Level	High		Medium		Low		Unspiked	
Microorganism	Pre-Hyb	Post-Hyb	Pre-Hyb	Post-Hyb	Pre-Hyb	Post-Hyb	Pre-Hyb	Post-Hyb
*G. vaginalis* ^1^	7.18 (0.85)	7.58 (1.36)	6.25 (0.72)	6.87 (1.17)	5.22 (0.61)	5.54 (0.99)	0.00 (0.00)	0.00 (0.00)
*L. crispatus* ^1^	7.20 (0.85)	7.49 (1.35)	6.26 (0.72)	6.81 (1.16)	5.28 (0.61)	5.45 (0.98)	0.00 (0.00)	0.00 (0.00)
*L. gasseri* ^1^	7.36 (0.87)	8.29 (1.49)	6.34 (0.73)	7.56 (1.28)	5.37 (0.62)	6.33 (1.14)	0.00 (0.00)	0.00 (0.00)
*A. vaginae* ^1^	7.24 (0.86)	7.46 (1.34)	6.36 (0.73)	6.77 (1.15)	5.39 (0.63)	5.49 (0.99)	0.00 (0.00)	0.00 (0.00)
*L. iners* ^1^	7.11 (0.84)	7.42 (1.33)	6.21 (0.72)	6.59 (1.12)	5.28 (0.61)	5.45 (0.98)	0.00 (0.00)	0.00 (0.00)
*P. agglomerans* ^2^	4.98 (0.59)	6.43 (1.16)	5.09 (0.59)	6.78 (1.15)	5.04 (0.58)	6.44 (1.16)	5.20 (0.61)	6.51 (1.13)
*Alternaria* sp. ^2^	5.36 (0.63)	4.76 (0.86)	5.55 (0.64)	5.19 (0.88)	5.55 (0.64)	4.96 (0.89)	5.61 (0.65)	5.12 (0.89)
16S ^2^	8.46	5.56	8.67	5.89	8.62	5.56	8.59	5.74

^1^ exogenous targets (vaginal bacteria *cpn60* plasmids). ^2^ endogenous targets (seed wash microorganisms).

**Table 3 microorganisms-09-00816-t003:** Correlations between *cpn60* gene copies determined by species-specific quantitative PCR and the number of sequencing reads mapping to each taxonomic cluster by CaptureSeq or amplification for 5 bacteria from a synthetic community spiked into a background of wheat seed washes. Samples were spiked at 4 levels (high, medium, low, unspiked) using *cpn60* plasmids from bacteria exogenous to the seed washes and quantified using qPCR and by read numbers determined from CaptureSeq and marker amplification. Significant correlations (*p* < 0.01) are shown in bold.

	Pearson (r^2^)		
Organism	CaptureSeq	Amplification	*n*
*Gardnerella vaginalis*	**0.999**	0.724	4
*Lactobacillus iners*	**0.998**	0.926	4
*Lactobacillus crispatus*	**0.998**	0.922	4
*Lactobacillus gasseri*	**1.000**	0.937	4
*Atopobium vaginae*	**1.000**	0.782	4
	**Spearman (ρ)**		
All combined	**0.956**	**0.912**	20

**Table 4 microorganisms-09-00816-t004:** Abundances of selected OTU from each domain in antibiotic treated soil samples, as determined by quantitative PCR.

OTU	Domain	cpnDB Nearest Neighbor	OTU Length (bp)	Sequence Identity (%) ^1^	Treatment (mg kg^−1^)	Soil Extract (Copies/g Soil)	Post-Hybridization Sample (Copies/µL)
XP002901426 DN2_c0_g1_i1	Eukarya (type I) ^2^	*Phytophthora infestans*	539	100	0	ND ^3^	1242
					10	ND	3942
WP036300323 DN4_c3_g1_i2	Bacteria (type I)	*Microbacterium lacus* C448	1066	99	0	6750	1417
					10	38,571 ^4^	8170 ^4^
KUL05486 DN0_c0_g1_i1	Archaea (type II)	*Methanoculleus marisnigri*	1029	92	0	495	ND
					10	527	3360

^1^ Percent identity to reference sequence in cpnDB. ^2^ Type I refers to the ~60 kDa mitochondrial and chloroplast proteins found in Bacteria, Eukarya, and certain Archaea. Type II refers to TCP1, the cytoplasmic orthologue of the group I chaperonins found in Archaea. ^3^ ND, not detected. ^4^ Statistically significant difference (*p* < 0.01) between 0 mg kg^−1^ and 10 mg kg^−1^ groups, using a Mann–Whitney rank sum test.

## Data Availability

DNA sequencing data associated with this work has been deposited at NCBI under BioProject PRJNA406970 (https://www.ncbi.nlm.nih.gov/bioproject/PRJNA406970, accessed on 1 April 2021) and SRA deposits SRX3181274-SRX3181276 and SRX3187583-SRX3187601 (https://www.ncbi.nlm.nih.gov/sra/, accessed on 1 April 2021).

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
