# Peer review of "CaptureSeq: Hybridization-Based Enrichment of cpn60 Gene Fragments Reveals the Community Structures of Synthetic and Natural Microbial Ecosystems"

_microorganisms, 2021, doi:10.3390/microorganisms9040816_

Round 1

Reviewer 1 Report

Links et al discuss a novel sequencing approach combining existing sequence capture technology with the universal chaperon 60 (cpn60) gene. Current popular microbiome sequence techniques involve amplicon or metagenomic approaches which have inherent biases associated with them (amplification or population biases). CaptureSeq aims to eliminate these biases by pulling down the respective cpn60 sequences across all taxa of a particular microbiome, eliminating the need for extensive application. Overall this group saw that the CaptureSeq method resulted in a better representation of the taxonomic distribution of different microbiomes when compared to existing sequencing methods.

Minor changes:

  • Line 176: Can you add the PCR conditions for the 16S primers, since you added them for the cpn60 UT.
  • Line 177: Why were their 40 cycles of PCR done, this can conflate the amplification bias. Was this PCR protocol tested for fewer cycles (like 25-30 cycles)?
  • Line 191: Can you be more specific on which prep kit you used, there are multiple options of this kit (ie multiplex Oligos or the NEBNEXT ultra).
  • Line 191: Was this kit amplification free or were there cycles of PCR done with this kit. Some Illumina kits (TrueSeq library prep) are amplification free
  • Line 290: I would remove this statement or include this data in the supplement.
  • Figure 1:  Personally I feel like a box plot or a non-stacked bar chart with error bars would illustrate this data better. It is hard to track the different genera with a stacked bar chart.
  • Figure 4 and Figure 5: I know you mention it in the figure label but there should be a color legend in the figure itself.
  • Figure 7: This should also have a legend for your shapes and colors.
  • The page numbers are off around page 8 and 9, could just be an error in the PDF creation

Major changes:

  • Is it possible to include a cost table in this manuscript that looks at the cost breakdown between the different sequencing done for this study? 
  • Also, I recommend adding more background on sequence capture technology to the introduction as I don't know how popular this technique is. This would also include explaining the difference between an OTU and an sOTU (I feel like this was a bit confusing in the manuscript as is) 

Overall I really enjoyed this study and I think it has the potential to introduce a new sequencing option for microbiome researchers. Once the concerns are addressed I would have no problem recommending this for publication. 

Author Response

Links et al discuss a novel sequencing approach combining existing sequence capture technology with the universal chaperon 60 (cpn60) gene. Current popular microbiome sequence techniques involve amplicon or metagenomic approaches which have inherent biases associated with them (amplification or population biases). CaptureSeq aims to eliminate these biases by pulling down the respective cpn60 sequences across all taxa of a particular microbiome, eliminating the need for extensive application. Overall this group saw that the CaptureSeq method resulted in a better representation of the taxonomic distribution of different microbiomes when compared to existing sequencing methods.

  1. Line 176: Can you add the PCR conditions for the 16S primers, since you added them for the cpn60 UT.

Response: Added (lines 195-196).

  1. Line 177: Why were their 40 cycles of PCR done, this can conflate the amplification bias. Was this PCR protocol tested for fewer cycles (like 25-30 cycles)?

Response: We have clarified that we followed the standard protocol for preparing cpn60 UT amplicon libraries for sequencing on the MiSeq by citing the recently provided protocol (Hill and Fernando Nat Protocol Exch 10.21203/rs.3.pex-1438/v1, 2021). This is now reference 45 (line 198).

  1. Line 191: Can you be more specific on which prep kit you used, there are multiple options of this kit (ie multiplex Oligos or the NEBNEXT ultra).

Response: We have indicated that we used the NEBNext Ultra sequencing prep kit (line 211).

  1. Line 191: Was this kit amplification free or were there cycles of PCR done with this kit. Some Illumina kits (TrueSeq library prep) are amplification free

Response: We appreciate that the reviewer has pointed this out. In fact there are limited PCR cycles done (6-12 using adaptor-specific primers) in preparing samples for sequencing. This has been clarified (line 212-213).

  1. Line 290: I would remove this statement or include this data in the supplement.

Response: This text has been removed – we feel that the revised version of Figure 1 makes the point suitably well.

  1. Figure 1:  Personally I feel like a box plot or a non-stacked bar chart with error bars would illustrate this data better. It is hard to track the different genera with a stacked bar chart.

Response: Figure 1 has been replaced with a box plot as suggested. We thank the reviewer for this suggestion and we feel that the revised figure is an improvement that conveys both the representation and the variability observed in this experiment.

  1. Figure 4 and Figure 5: I know you mention it in the figure label but there should be a color legend in the figure itself.

Response: Added (Figure 4, Figure 5)

  1. Figure 7: This should also have a legend for your shapes and colors.

Response: Added (Figure 7).

  1. The page numbers are off around page 8 and 9, could just be an error in the PDF creation

Response: Fixed (page headers throughout document)

Major changes:

  1. Is it possible to include a cost table in this manuscript that looks at the cost breakdown between the different sequencing done for this study?

Response: We have included a new supplemental table (Table S4) that contains cost information (lines 488-491).

This one will be a little difficult; cost for CS will depend on scale of synthesis and of course country (provide in US dollars I suppose). Will ask our contacts at Illumina for some advice. Then I have to look up the pricing for the Arbor Biosciences probes – maybe provide various synthesis formats and provide per-hyb prices or something. Will be an estimate. Supplemental table perhaps.

  1. Also, I recommend adding more background on sequence capture technology to the introduction as I don't know how popular this technique is. This would also include explaining the difference between an OTU and an sOTU (I feel like this was a bit confusing in the manuscript as is)

Response: We added some more background on OTU/sub-OTU definition (lines 108-114), and expanded on the use of sequence capture to enrich environmental DNA samples for sequences of interest (lines 122-127).

Overall I really enjoyed this study and I think it has the potential to introduce a new sequencing option for microbiome researchers. Once the concerns are addressed I would have no problem recommending this for publication. 

Response: We appreciate the positive comments!

Reviewer 2 Report

Authors present a study on the development of hybridization-based enrichment of cpn60 gene fragments to characterize microbiota. Validation of the strategy is conducted though MOCK microbial communities then applied to various environmental samples. The technique is compared to currently used profiling methods such as 16S metabarcoding and shotgun metagenomics. Gene capture by hybridization is an innovative approach avoiding various biases linked to current protocols. Simultaneously Bacterial, Archaeal and Eukaryal diversity could be explored. The results are promising even if no comparison have been made with 16S rRNA gene capture by hybridization.

Major comments :

Richness and diversity comparison used very few reads to obtain a complete description of the diversity. It is important to explain why authors used so little depth sequencing for metagenomics approach and to be caution for the interpretations. Furthermore, one of the major goal in microbial profiling is to be able to describe the communities at species level. This is a major bias of metabarcoding.

Authors made first comparison at species-level with 16S metabarcoding that is most of the time impossible due to short-targeted regions. Nearly perfect results are obtained using mapping on reference genomes but it will be important to make comparisons without such knowledge. Authors showed that few amplification biases (depending on GC content) on a simplified microbial community. To my opinion, the interesting result is that cpn60 capture by hybridization is efficient. Reconstruction of full length genes allows more precise affiliation.

The quantification in Table 2 using spiked cpn60 genes before and after hybridization is a little bit confusing and should be clearer for the readers. Author claim enrichment 2-3 orders magnitude for cpn60-DNA fragments than to 16S rRNA genes (lanes 333-336). How is it possible to obtain such conclusion? 16S experiment indicates that this marker is not targeted. Why no enrichment is observed comparing pre-hyb and post-hyb for spiked cpn60 DNA fragments? Table 2 and associated description in the text must be improved. What is important is to determine the efficiency of gene capture by hybridization. Same comment for Table 3 and the cpn60 amplifications (marker amplification). It is also confusing for the readers. The cpn60 amplification approach appears here but not previously. It is an important result showing that hybridization capture kept real abundances.

For natural ecosystems exploration, what could be the affiliation precision with capture by hybridization (… genus, species level)? It is important to precise that point. Diversity indexes should be manipulated with caution due to very low depth sequencing. Do the Authors have example with higher sequencing depth?

In the discussion part, Authors well presented the limits and possible improvements. What are the limits regarding cpn60 and 16S rRNA gene databases?

Minor comments :

Lane 419 : Anopheles?

Author Response

Authors present a study on the development of hybridization-based enrichment of cpn60 gene fragments to characterize microbiota. Validation of the strategy is conducted though MOCK microbial communities then applied to various environmental samples. The technique is compared to currently used profiling methods such as 16S metabarcoding and shotgun metagenomics. Gene capture by hybridization is an innovative approach avoiding various biases linked to current protocols. Simultaneously Bacterial, Archaeal and Eukaryal diversity could be explored. The results are promising even if no comparison have been made with 16S rRNA gene capture by hybridization.

Major comments :

  1. Richness and diversity comparison used very few reads to obtain a complete description of the diversity. It is important to explain why authors used so little depth sequencing for metagenomics approach and to be caution for the interpretations. Furthermore, one of the major goal in microbial profiling is to be able to describe the communities at species level. This is a major bias of metabarcoding.

Response: We agree that the alpha diversity metrics (Figure 5) were calculated on a low number of cpn60 reads as determined by the lowest-yielding method, which was shotgun genomics. The point of this Figure was not to measure with accuracy these metrics in the soil samples, but to compare (at equivalent sampling depths) the values of these metrics, along with the variability among biological replicates observed using the three profiling methods. We have added some text to clarify this point, and to emphasize that our goal was not to obtain an accurate measure of the alpha diversity of the soil samples (lines 498-503).

Regarding the discrimination of microbial taxa at the species level, we certainly agree – and we point out that one of the strengths of the cpn60 universal target as a taxonomic marker is this very ability to discriminate below the genus level. For example, pairwise identities of cpn60 UT sequences between pairs of species accurately predict whole-genome sequence identities, and hence species affiliations (Verbeke et al. Sys Appl Microbiol 2011, 34:171-179). In addition, cpn60 sequences have been used to discriminate subspecies of Gardnerella vaginalis (Paramel Jayaprakash et al. PLoS ONE 2012, 7(8):e43009), and have been shown to be a suitable barcode target for species and pathovar-level identification of plant pathogenic Xanthomonas spp. (Tian et al. PLoS One 2016, 11(11):e0165995). We added some text in the Introduction to highlight this point (lines 101-108).

  1. Authors made first comparison at species-level with 16S metabarcoding that is most of the time impossible due to short-targeted regions. Nearly perfect results are obtained using mapping on reference genomes but it will be important to make comparisons without such knowledge. Authors showed that few amplification biases (depending on GC content) on a simplified microbial community. To my opinion, the interesting result is that cpn60 capture by hybridization is efficient. Reconstruction of full length genes allows more precise affiliation.

Response: We agree that, in unknown environments, the attribution of 16S rRNA-encoding sequences to taxonomic levels below genus is often difficult. It was possible to ascribe 16S sequences to the species level in the zymobiomics panel due to the known composition. One of the features of the cpn60 taxonomic target is its ability to discriminate at the species level or even below (Hill JE, Vancuren SJ. Database 2019, 2019). This improved discrimination of bacterial taxa compared to 16S allowed us to demonstrate that cpn60 meets the criteria of a barcode marker for Bacteria, as it features a wider barcode gap (Links et al. PLoS One 2012, 7(11):e49755.). We agree that longer sequences provide improved taxonomic classification. Some relevant text and references have been added to the discussion to highlight this point (lines 627-630).

  1. The quantification in Table 2 using spiked cpn60genes before and after hybridization is a little bit confusing and should be clearer for the readers. Author claim enrichment 2-3 orders magnitude for cpn60-DNA fragments than to 16S rRNA genes (lanes 333-336). How is it possible to obtain such conclusion? 16S experiment indicates that this marker is not targeted. Why no enrichment is observed comparing pre-hyb and post-hyb for spiked cpn60 DNA fragments? Table 2 and associated description in the text must be improved. What is important is to determine the efficiency of gene capture by hybridization. Same comment for Table 3 and the cpn60 amplifications (marker amplification). It is also confusing for the readers. The cpn60 amplification approach appears here but not previously. It is an important result showing that hybridization capture kept real abundances.

Response: We based this conclusion on the fact that the apparent abundances of 16S genes decreased approximately 1000-fold after hybridization, as determined by qPCR targeting global 16S genes. For example, 16S genes in the “high” spike level were log10 8.46 before hybridization, and log10 5.56 after – a difference of log10 2.9, or 794. Similarly, in the “low” spike level the 16S genes decreased 1148-fold (8.62-5.56=3.06). As 16S genes were not targeted by the probes, they were expected to decrease after hybridization. We suspect that even better depletion of nontarget genes would be obtained with more recent washing protocols that resulted in better enrichment of cpn60 genes (lines 598-601) – still not close to the levels of enrichment observed by Gasc and Peyret for 16S, but an improvement nevertheless. Since cpn60 genes were the target of the probes, we expected that their abundances would not change dramatically (i.e., they would be retained and preserved through the washing procedure) after hybridization. We have made changes to Table 2 (added ratios of cpn60/16S genes before and after hybridization) to help clarify these points, and have added some text that we hope will clarify (lines 354-359).

  1. For natural ecosystems exploration, what could be the affiliation precision with capture by hybridization (… genus, species level)? It is important to precise that point. Diversity indexes should be manipulated with caution due to very low depth sequencing. Do the Authors have example with higher sequencing depth?

Response: Examination of cpn60 sequences will provide higher resolution discrimination of microbial taxa (below the Genus level) compared with 16S rRNA gene profiling. This feature is not different among amplification, capture-seq, or shotgun profiling. We trust that the text we have added to the Introduction will orient readers to the advantages of cpn60-based microbial community profiling (lines 101-108). We agree that the alpha diversity indices were calculated based on a low number of reads, and have clarified that we cannot draw conclusions regarding the amplitude of these indices in the soil samples as a result (lines 510-513). For this comparative work, we performed the shotgun sequencing on a MiSeq instrument for maximum consistency – this does not provide the depth required for typical shotgun analysis, which would use other platforms such as HiSeq or NovaSeq. We felt it was important to be as consistent as possible among the methods compared, while acknowledging that the least biased method (which also provides additional information on metabolic capacity) is shotgun sequencing (lines 84-85).

  1. In the discussion part, Authors well presented the limits and possible improvements. What are the limits regarding cpn60and 16S rRNA gene databases?

Response: The publicly available cpn60 reference database provides nearly equivalent taxonomic breadth compared to 16S rRNA gene databases (Hill JE, Vancuren SJ. Database 2019, 2019), although clearly not the depth since it is not as widely used. As for the reference database used in the present work for mapping and assembly, we showed in Supplemental Dataset S2 that the sequences retrieved from GenBank provided equivalent taxonomic breadth to the main 16S and ITS databases in use (lines 230-232). We added some text in the Introduction to clarify this point (lines 132-134).

Minor comments :

  1. Lane 419 : Anopheles?

Response: Thanks to the reviewer for pointing out this error. We have corrected this (line 437).

Reviewer 3 Report

This is an impressive study with many sophisticated and modern methods. The results obtained may be important for applied science and in general could have significance in the science development in very important area associated with microbiome studies.  The manuscript is very well prepared. Different methods for microbiome studies are described with their positive sides and shortcomings.

Some minor suggestions from my side:

  1. rows 167-168. Please describe a bit in more detail how does biological material was recovered after centrifugation (which part was taken from the liquid for  further DNA extraction).
  2. Please try to make a bit larger legend in the Figure 1 (the columns can be a bit more narrow if necessary).
  3. Please check the page numbers continuing through all document as it is something wrong with them at present.

Author Response

This is an impressive study with many sophisticated and modern methods. The results obtained may be important for applied science and in general could have significance in the science development in very important area associated with microbiome studies.  The manuscript is very well prepared. Different methods for microbiome studies are described with their positive sides and shortcomings.

Some minor suggestions from my side:

  1. rows 167-168. Please describe a bit in more detail how does biological material was recovered after centrifugation (which part was taken from the liquid for  further DNA extraction).

Response: We clarified that DNA was extracted from the pellet generated from centrifugation of 2L of water (line 185).

  1. Please try to make a bit larger legend in the Figure 1 (the columns can be a bit more narrow if necessary).

Response: We have replaced Figure 1 with a new version according to the suggestion of reviewer 1. We feel that the readability of the figure has been improved.

  1. Please check the page numbers continuing through all document as it is something wrong with them at present.

Response: Fixed (page headers throughout document)

Round 2

Reviewer 2 Report

The Authors well answered to comments.

The revised form is now adapted for publication.